# An Investigation of the Effects of Additives and Burning Temperature on the Properties of Products Based on Loam

Ruslan E. Nurlybayev [1,*], Maratbek T. Zhuginissov [2], Zhanar O. Zhumadilova [2,*], Aidos A. Joldassov [2], Yelzhan S. Orynbekov [2] and Aktota A. Murzagulova [1]

1   LLP SAVENERGY, Industrial Zone, Almaty A03A2C6, Kazakhstan; murzagulova.savenergy@gmail.com
2   Institute of Architecture and Civil Engineering, Satbayev University, 22a Satpaev St., Almaty A03A2C6, Kazakhstan; m.zhuginissov@satbayev.university (M.T.Z.); a.joldassov@satbayev.university (A.A.J.); orynbekov.savenergy@gmail.com (Y.S.O.)
*   Correspondence: nurlybayev.savenergy@gmail.com (R.E.N.); z.zhumadilova@satbayev.university (Z.O.Z.); Tel.: +7-777-138-86-77 (Z.O.Z.); Fax: +7-727-292-60-25 (Z.O.Z.)

**Abstract:** This study aimed to identify the composition of ceramic mass for the manufacture of bricks with improved properties based on local loam using diatomite and bentonite clay. For the experiments, loam from the Almaty deposit (Kazakhstan) was applied as the primary product, and used for the production of ceramic bricks with grades of 75 and 100. Diatomite from the Zhalpak and Utesai deposits and highly plastic bentonite clay from the Darbazin deposit (Kazakhstan) were used as additives. An analysis of the properties of the burned products demonstrated that supplementation with diatomite and bentonite in the loam lowers the average density and increases the compressive strength of samples burned at 1000 and 1100 °C. Herewith, the optimal amount of diatomite and bentonite clay to add is 15%, with a fractional composition of 0.315–0.16 mm and less than 1 mm, respectively. It was established that the optimal additions of diatomite and bentonite clay to loam make it possible to obtain after burning at 1170 °C samples of ceramic products corresponding to the 2nd class in terms of the average density and compressive strength grades M 400 and M 500.

**Keywords:** loam; diatomite; bentonite clay; burning; density; strength; clinker brick

## 1. Introduction

Brick has been used in construction since time immemorial, as evidenced by many historical sources. In fact, it was the first building module, that is, a unified product, from "grass" material, which made it possible to solve many structural and architectural problems. However, even today, brick has not lost its relevance. It is constantly evolving, and improving the previous properties. Years have no power over brick as houses built from it have never become obsolete, and only acquire a noble stamp of time. At the same time, the material has many advantages. It is strong, durable, stable, and versatile. It has a good ability regarding air exchange; that is, the brick breathes and allows the inhabitants of the house to breathe. Brick walls heat up slowly and cool down slowly, keeping the inhabitants warm in winter and cool in summer. Finally, they absorb excess moisture, after which they give up their water "reserves" to the air, having a beneficial effect on the microclimate of the house.

To expand the resource base and improve the technical performance, research is being conducted on the application of native and man-made raw materials as additives in the composition and technology of ceramic wall products.

The authors of [1] used granulated phosphorus slag from Nodfos JSC (Taraz, Kazakhstan) to develop the compositions and technology of wall ceramics. Plastic clay, loam, and shale were used as plasticizing additives. To increase the porosity of the brick, ash and coal dust were added to the ceramic mass. The content of slag in the ceramic mass was

30–80 wt.%. The resulting bricks had an average density ranging from 1608 to 1870 kg/m$^3$ and a compressive strength ranging from 17.4 to 36.1 MPa.

At Dulaty Taraz Regional University carried out a study on the development of energy- and resource-efficient ceramic bricks based on blast-furnace slags of the Karaganda Metallurgical Plant JSC "Ispat-Karmet" [2]. The compositions of ceramic bricks containing 67 to 75% granulated blast furnace slag were proposed. Bentonite clay, glass slag, and alkaline additive in the form of soda were used as binder components. The compressive strength of the obtained bricks was 17 to 19 MPa, which corresponds to grades M 150 to M 200.

The authors of [3] used granulated phosphorus slag in ceramic brick technology. Before making up the charge, the granulated phosphate slag was ground together with bentonite in a ball mill until it passed through a 0.315 sieve. The ratio of phosphate slag to bentonite was 4:1.67 by weight. The ceramic mass included raw (primary) components in the following ratios: wt.%:granulated phosphate slag, from 40 to 60; loam, from 25 to 40; and bentonite clay, from 15 to 20. This ceramic mass had a plasticity number (P) of 9.5 to 11 and showed little sensitivity to drying ($K_i$ = 0.50–0.70). The compressive strength of the ceramic bricks ranged from 17.2 to 21.5 MPa.

In the patent for the invention [4], to reduce the average density of ceramic bricks, a ceramic mass was developed that included (% of mass) 55–65 easy-melting clay and 35–45 carbonate-silica rock with a ratio of calcium oxides to silicon of 0.31–0.34. The obtained bricks showed an average density in the range of 1490–1570 kg/m$^3$, porosity of 38.4 to 41.9%, and compressive strength of 18.6 to 27.5 MPa.

In [5], for the manufacturing of hollow core porous products, a study on the easy-melting clay Atratyev deposit (facility) with carbonate trepel Novo-Aibesinov deposit (ratio 60:40) found that burning resulted in new crystalline formation, in the form of wollastonite, and a higher proportion of the glass phase, which was confirmed by its high compressive strength and flexural strength. The obtained corpulent lightweight material was M 250 grade with an average density of 1260 kg/m$^3$ and the hollow lightweight material was M 175 grade (mark) with a density of 800 kg/m$^3$ with water absorption (absorbency) of 20%. The capability to obtain clinker wall products at 1000–1170 °C from easy-melting clay from the Alekseev deposit with diatomite from the Inzen deposit (the ratio of clay and diatomite is 70:30) has also been established. The composition of feldspars and quartz decreases, the content of cristobalite increases significantly, and the fraction of the glass phase increases significantly, accompanied by an increase in the compressive strength of the samples from 80 to 120 MPa and the flexural strength from 27 to 50 MPa.

The authors of [6] proposed technology for the manufacturing of ceramic wall productions from a raw mixture consisting of 65% high-calcium fly ash, 35% microsilica, and 32% tall pitch emulsion (over 100%) with the addition of sodium hypochlorite at 5% (from weight) pitch. Lightweight ceramic products have been produced that meet the requirements of GOST 530–95 (GOST-Government standard) for wall ceramic products with an average density of 1230 kg/m$^3$, strength-grade M 100, frost resistance F 25, and thermal conductivity of 0.57 W/(m °C).

In [7], to reduce the density of ceramic bricks, tuff wastes with particle sizes less than 1 mm were added to raw clay materials. Ceramic products with an average density ranging from 1550 to 1700 kg/m$^3$, compressive strength of 15.2 to 24.3 MPa, and water absorption of 9 to 12.2% were obtained. Further, volcanic ash with a particle size of 1 mm was introduced into the raw clay material with montmorillonite and polymineral compositions. The resulting ceramic samples had a density of 1710 (1700) kg/m$^3$, compressive strength of 22.1 (15.8) MPa, and water absorption of 9.8 (13.9)%, respectively. Compared to the original samples (without ash), their density was reduced by 18%. The test results showed that volcanic ash and volcanic tuff waste are effective processing aids for clay in the production of ceramic wall materials with enhanced thermal insulation characteristics.

In [8], the ability to produce a lightweight aggregate based on loam from the Chagan field in the range of 50 to 80 wt.% and bentonite clay Pogadaev deposit (West Kazakhstan

region) in the range of 20 to 50 wt.% was investigated. Raw materials were mixed, moistened, and pelletized after preparation. Burning at 1150 °C for 20–30 min was conducted. The resulting lightweight aggregate showed the following characteristics: bulk density varied from 550 to 870 kg/m$^3$ and grade under compression in the cylinder 50–150 MPa. It has been established that the addition of bentonite to loess-like loam makes it possible to transfer loams from the non-intumescent category to the medium intumescent clay category. In addition, the presence of loess-like loam during the production of claydite increased the strength of the granules from 2.5 to 3 times.

The authors of [9,10] present the results of studies on technology for the application of lightweight aggregates to natural diatomite. Aggregates were prepared using clinker and burning technologies. In the clinker technology, Portland cement M 400 was applied as a binder. Light aggregate granules were hardened in wet conditions. In the burning technology, local loam, sawdust, and coal were used as additives. Aggregates with a bulk density of 610 to 627 kg/m$^3$ and a compressive strength in the cylinder of 2.82 to 3.84 MPa were produced using clinker technology. According to the burning technology, aggregates with a bulk density of 532 to 819 kg/m$^3$ and a compressive strength in the cylinder ranging from 12.5 to 19.4 MPa were obtained. The resulting lightweight aggregates met the requirements of GOST (government standard) 32496–2013 "Porous Aggregates for Lightweight Concrete. Technical regulations".

In [11], the development mass compounds based on local loam for the production of conditionally effective ceramics was studied. To reduce the average density of effective wall materials, crushed local reeds, fly ash formed from the combustion of coal under domestic conditions, and natural diatomite were applied as additives. From 5 to 20% of the ceramic weight of additives were introduced into the compound. in the amount The cylinder specimens were burned in a muffle furnace at temperatures of 950, 1000, and 1100 °C. With the addition of household slag, ceramic products were obtained with an average density from 1385 to 1410 kg/m$^3$ and a compressive strength ranging from 9.2 to 11.2 MPa. With the use of diatomite, products with an average density of 1438 to 1510 kg/m$^3$ and a compressive strength of 8.8 to 10.7 MPa were produced. In terms of density, the products correspond to conditionally effective and strength grades of M 75 and M 100.

In [12], the replacement of clay with 6% hot-dip galvanizing slurry and burning at 1020 °C showed admissible mechanical and ceramic parameters that met the requirements (demands) of Euro Code. When up to 20% sludge (collected from industrial wastewater treatment plants) was added to bricks, the strength measured at 960 and 1000 °C met the requirements of the Chinese National Standard, and replacing clay with 25% glass waste sludge and burning at 850 °C showed an improvement in the strength by compression of 37%. Replacing the clay with 10% ceramic mud and burning at 1050 °C satisfied the aesthetic requirements and retained sufficiently high mechanical properties (characteristics). Different types of sludge were mixed with different metals and minerals to form clay mixtures, which affected the properties of the burned clay bricks in different ways.

In [13], the impact of porosity is limited by the converting mineral of the material. The reaction between clay minerals and calcium carbonate results in the formation of new highly conductive mineral phases, such as gelenite (1.53 W/m·K) and wollastonite (4.03 W/m·K). After exposure to 850 °C, compaction and elimination of pores occurred in the brick samples, which led to an increase in the thermal conductivity.

In [14], the results of the use of wastewater treatment technologies in soil-cement bricks showed that this supports a lower impact of wastewater treatment technologies on the environment. The application of spent formed sand along with gravel dust reduced water absorption and provided an acceptable level of mechanical strength corresponding to the established norms for soil-cement bricks. The possibilities of recycling glass for the production of soil-cement bricks were also considered.

In [15], when considering raw and finished ceramic materials using various analytical methods, it was found that changes in the physical properties are associated with the

formation of glassy structures and new crystalline minerals during sintering at temperatures ranging from 700 to 1100 °C. Red clay with ground easy-melting cullet and biowaste for brick manufacturing at sintering temperatures ranging from 959 to 1000 °C showed the physical properties depend on the waste content. Waste ceramics were used as a substitute for clay and paper pulp in various weight percentages from 5 to 15% to produce raw bricks for the analysis of thermal and physical data. A decrease in the thermal conductivity from 0.75 to 0.58 W/m·K was noted with an increase in the porosity from 3.5 to 35%.

Thus, at present, a significant amount of research has been carried out on compounds and technology of ceramic bricks, light aggregates based on primary and man-made raw materials, and mixtures containing both primary and man-made raw materials. Clay, shungite, silica-containing raw materials (diatomite, tripoli and flask, gas-forming additives), finely ground limestone, sawdust, buckwheat husk, and tuff waste are used as natural raw material components. As technogenic raw materials, coal enrichment waste, ash and slag, and fly ashes are used.

The developed ceramic products, depending on the type of additive used and the aim, have a wide range of properties that meet the requirements of regulatory documents.

The goal of this study was to develop ceramic mass compounds for the manufacturing of bricks with improved properties of local loam based on the use of diatomite and bentonite clay.

Regarding the novelty of this work, it was found that diatomite and bentonite clay addition to loam increased the strength of the products while decreasing their average density due to the formation of a glass phase, which formed during the sintering of ceramic shards. It was found that diatomite additives led to the formation of larger quantities of augite, and the addition of bentonite clay increased the quartz content in the products. It was shown that products with augite have higher strength characteristics.

## 2. Materials and Methods

### 2.1. Materials

For the experimental work, loam from the Almaty deposit (Kaskelen region), located 4–5 km south of the existing brick plant, was applied as the basic source. The plasticity number of loam is 8.9 and it has lower plasticity; according to its fire resistance, it is an easy-melting raw clay product. Is is suitable for the manufacturing of frost-resistant bricks of grades 100 and 125 by plastic pressing, which meets the technical regulations of GOST 530–2012 "Brick and ceramic stones".

Diatomite of the Zhalpak and Utesai deposits (Aktobe region) and highly plastic bentonite clay of the Darbaz deposit (Turkestan region) were used as additives.

For the preparation of ceramic masses, loam and bentonite clay were used after crushing and sieving through a 1 mm sieve.

After milling and crushing, diatomite was applied until complete passage through a 0.315 mm sieve. To study the effect of the fractional compound of diatomite, it was crushed and sieved through 2.5, 1.25, 0.63, 0.315, and 0.16 sieves.

After dosing, the loam and additives were thoroughly mixed first in a dry state, and then water was added in the quantity required to obtain a plastic mouldable mass. To study the product characteristics, standard specimens (cylinders) were prepared with a diameter of 50 mm and a height of 50–55 mm. The sample cylinders were formed operating a hydraulic press at a pressure of 2–4 kN.

The samples were dried in a drying cabinet DC 80–01 SPU oven at a temperature of 95–100 °C for 1–2 h. After drying, the samples' mass and dimensions were determined. The samples were burned at temperatures of 1000, 1100, 1150, 1170, and 1200 °C in a muffle furnace SNOL 1.6/1300. The temperature speed rate in the furnace was 5 °C/min, and the final temperature was maintainedfor 1 h.

*2.2. Methods*

2.2.1. Methods for Studying the Structure and Phase Transformations

X-ray diffraction analysis was carried out on a DRON-3.0 diffract meter with $Cu_{K\alpha}$-radiation and $\beta$-filter. Th conditions used to obtain the diffractograms were as follows: U = 35 kV; I = 20 mA; shooting θ–2θ; detector 2 deg/min. X-ray phase analysis on a semi-quantitative basis was performed based on powder samples' diffraction patterns using the method of equal mass and artificial mixtures. The quantitative ratios of the crystalline phases were determined. Interpretation of the diffraction patterns was carried out using the data from the ICDD file cabinet: the PDF2 (Powder Diffraction File) database of powder diffraction data and the diffraction patterns of minerals free of impurities.

Analysis of the elemental compound of the materials and images using various types of radiation were performed with an INCA ENERGY energy-dispersive spectrometer (OXFORD INSTRUMENTS, England) installed on a Super probe 733 electron probe micro analyzer at an accelerating electrical voltage of 25 kV and a probe current of 25 nA.

2.2.2. Methods for Studying the Physical and Mechanical Properties of the Products

The average density of the samples was determined on the sample -cylinders with a diameter of 50 mm and height of 50 mm. The average density of the products was determined and calculated according to the methodology [16].

The strength (kg·f/cm$^2$, MPa) was determined on the sample cylinders with a diameter of 50 and a height of 50 mm in accordance with [16,17].

Determination of the total shrinkage was carried out on the sample cylinders with a diameter of 50 mm and a height of 50 mm. The total volume shrinkage is the sum of the air volume shrinkage and fire volume shrinkage. The air volume shrinkage was determined and calculated according to the method of [18,19].

## 3. Results and Discussion

This section presents the main results and discussions of the study of the compounds and technological parameters required to obtain ceramic samples for the production of bricks with optimal characteristics (properties) based on local loam using diatomite and bentonite clay. This is particularly relevant to the production of ceramic products that can be used for erecting the walls of residential buildings and exterior decoration. The effect of each additive was studied separately.

*3.1. Study of the Diatomite Influence from Zhalpak Deposit on the Properties of Products*

Table 1 presents some results from the determination of the properties of materials including pure loam and diatomite from the Zhalpak deposit.

After burning at 1000 °C, the pure loam sample had a density of 1601 kg/m$^3$ and a compressive strength of 10 MPa. After burning at 1100 °C, the pure loam sample had a density of 1647 kg/m$^3$ and a compressive strength of 17.46 MPa.

The study of the influence of diatomite additives from the Zhalpak diatomite deposits showed that with an increase in the amount of diatomite in the sample compound, regardless of the burning temperature, in general, a reduction in the compressive strength and average density of the samples was observed. So, starting with the additive of 15% to 50% diatomite, the average density of the samples decreased from 1560 to 1283 kg/m$^3$. For the ceramic masses containing 20–30% diatomite, conditionally effective wall materials with an average density of 1410 to 1550 kg/m$^3$ and a compressive strength of 10 to 14 MPa were obtained.

When studying the effect of diatomite from the Zhalpak deposit on the properties of loam, it was found that the 15–20% addition increased the strength and a decreased the density of the products. Similar results were obtained in [11], where the addition of 10, 15, and 20% diatomite to loam allowed products corresponding to the density of the conditionally efficient and strength grades of M 75 and M 100 to be obtained. The average density of the products was reduced after the addition of diatomite due to its low bulk

density, which is, on average, 800 kg/m$^3$, and, in this case, the burning temperature at which there was incomplete sintering of the ceramic tiles.

**Table 1.** The compounds and properties of samples of ceramic materials with diatomite from the Zhalpak deposit.

| No. | Compositions (Compounds) | $\rho_{av.}$, kg/m$^3$ | | Compressive Strength, MPa | | Full (Total) Volumetric Shrinkage, % | |
|---|---|---|---|---|---|---|---|
| | | 1000 °C | 1100 °C | 1000 °C | 1100 °C | 1000 °C | 1100 °C |
| 1 | Loam-100% | 1601 | 1647 | 10.0 | 17.46 | 3.8 | 4.1 |
| 2 | Loam-85%, diatomite fr. <0.315 mm-15% | 1560 | 1565 | 14.76 | 14.90 | 4.5 | 4.6 |
| 3 | Loam-80%, diatomite fr. <0.315 mm-20% | 1538 | 1525 | 14.9 | 13.24 | 4.3 | 4.1 |
| 4 | Loam-70%, diatomite fr. <0.315 mm-30% | 1410 | 1439 | 13.3 | 13.27 | - | - |
| 5 | Loam-60%, diatomite fr. <0.315 mm-40% | 1377 | 1347 | 13.27 | 11.76 | - | - |
| 6 | Loam-50%, diatomite fr. <0.315 mm50% | 1283 | 1361 | 12.4 | 9.71 | - | - |

### 3.2. Study on the Influence of Diatomite from the Utesai Deposit on the Properties of Products

In this study, the impact of the fractional compound on the characteristics of the products was studied. Crushed diatomite powders from the Utesai deposit were used in the following fractions: 2.5–1.25, 1.25–0.63, 0.63–0.315, and 0.315–0.16 mm. These fractions were added to the compound to reach a ceramic weight of 15%. The samples were burned at temperatures of 1100 and 1150 °C for 1 h.

Table 2 presents the compounds and their determined characteristics after burning at 1100 and 1150 °C based on loam with diatomite with various fractional compositions.

**Table 2.** The influence of the fractional diatomite compound and firing temperature on the properties of the products.

| No. | Compounds | $\rho_{av.}$, kg/m$^3$ | | Compressive Strength, MPa | | Full (Total) Volumetric Shrinkage, % | |
|---|---|---|---|---|---|---|---|
| | | 1100 °C | 1150 °C | 1100 °C | 1150 °C | 1100 °C | 1150 °C |
| **1** | Loam-100% | 1647 | - | 17.46 | - | 4.1 | - |
| 2 | Loam-85%, diatomite fr. 2.5–1.25 mm-15% | 1401 | - | 3.5 | - | 4.4 | - |
| 3 | Loam-85%, diatomite fr. 1.25–0.63 mm-15% | 1447 | - | 6.6 | - | 4.6 | - |
| 4 | Loam-85%, diatomite fr. 0.63–0.315 mm-15% | 1488 | - | 12.2 | - | 4.2 | - |
| 5 | Loam-85%, diatomite fr. 0.315–0.16 mm-15% | 1521 | 1615 | 14.8 | 29.4 | 4.5 | 6.3 |

As shown in Table 2, the fractional composition of diatomite had a noticeable effect on the average density and compressive strength of the samples (specimens). So, the additive fractions of 2.5–1.25 and 1.25–0.63 mm significantly lowered the compressive strength of the samples compared with the samples of pure loam by 80% and 62.2%, respectively, despite the high burning temperature (1100 °C). The addition of 0.63–0.315 and 0.315–0.16 mm fractions reduced the strength of the samples by 30% and 15.2%, respectively. According to the requirements, theGOST grade strength of the ceramic samples is M 125 and M 150, and the average density allowed us to identify them asconditionally efficient products. Thus, the addition of diatomite fractions of 2.5–1.25 and 1.25–0.63 mm noticeably reduced the sintering of loam, which led to a significant decrease in the compressive strength of the products. Considering that the addition of diatomite fractions of 0.315–0.16 mm at 15% was the most optimal, samples with the № 5 composition were burned at a temperature of 1150 for 1 h after drying. The average properties of the samples burned at 1150 °C are shown in Table 3.

**Table 3.** Compounds and properties of samples based on loam and diatomite.

| No. | Compounds | $\rho_{av.}$, kg/m$^3$ | Compressive Strength, MPa | Full (Total) Volumetric Shrinkage, % |
|---|---|---|---|---|
| | | 1170 °C | 1170 °C | 1170 °C |
| **1** | Loam-100% | 1927 | 38.5 | 6.0 |
| 2 | Loam-85 %, diatomite fr. 0.315–0.16 mm-15% of the Zhalpak deposit | 1945 | 43.3 | 6.2 |
| 3 | Loam-85%, diatomite fr. 0.315–0.16 mm-15% of the Utesai deposit | 1944 | 45.9 | 6.5 |

Figure 1 shows photographs of the samples and their structure after the compressive strength tests.

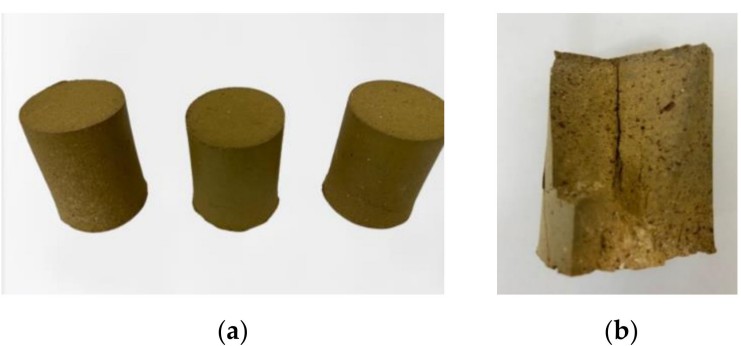

(**a**)                (**b**)

**Figure 1.** Samples with the addition of 0.315–0.16 mm diatomite fraction after burning at 1150 °C.

It can be seen that the samples had a baked dense shard. Despite the porous structure of the sample (Figure 1b), the compressive strength of the sample was 29.4 MPa at a density of 1615 kg/m$^3$, which corresponds to the density of the sample based on pure loam burned at 1100 °C. This high strength is due to the sintering process of the ceramic shard of the product. Such high compressive strength, according to GOST, is inherent in M 300 grade brick.

The next stage of this study was the selection of a burning temperature that results in the shard being in a fully sintered state;that is, obtaining a sample with the structure of a clinker brick. Using optimal amounts of diatomite fractions of 0.315–0.16 mm from the Zhalpak and Utesai deposits, samples were molded, which, after drying, were burned in a muffle furnace at 1170 and 1200 °C for 1 h.

As can be seen from Figure 2, after burning at a temperature of 1170 °C, the samples showed a shiny vitrified surface, which indicated the sintered state of the ceramic shard. The fracture structure of the sample with diatomite from the Utesai deposit was denser and more uniform.

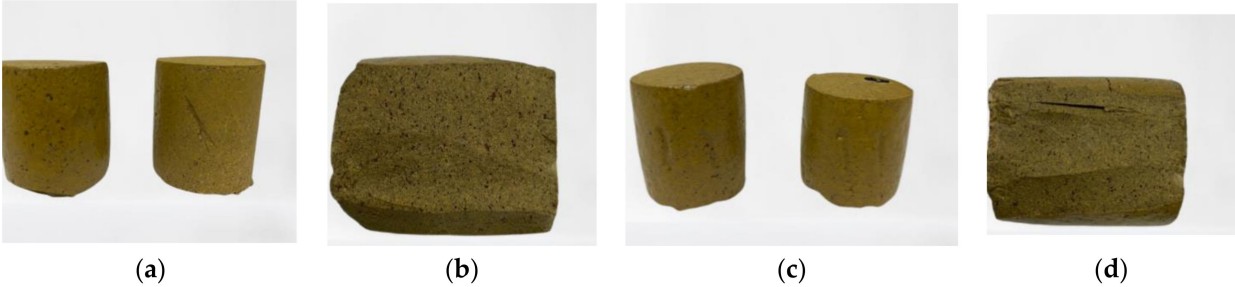

(**a**)  (**b**)  (**c**)  (**d**)

**Figure 2.** The samples with diatomite after burning at a temperature of 1170 °C: (**a**) samples with the addition of diatomite from the Zhalpak deposit. (**b**) Sample structure after the compression test. (**c**) Samples with the addition of diatomite from the Utesai deposit. (**d**) Structure of the sample after the compression test.

Table 3 shows the physical and mechanical properties of the samples after testing. Burning at 1200 °C led to complete deformation of the specimens (Figure 3).

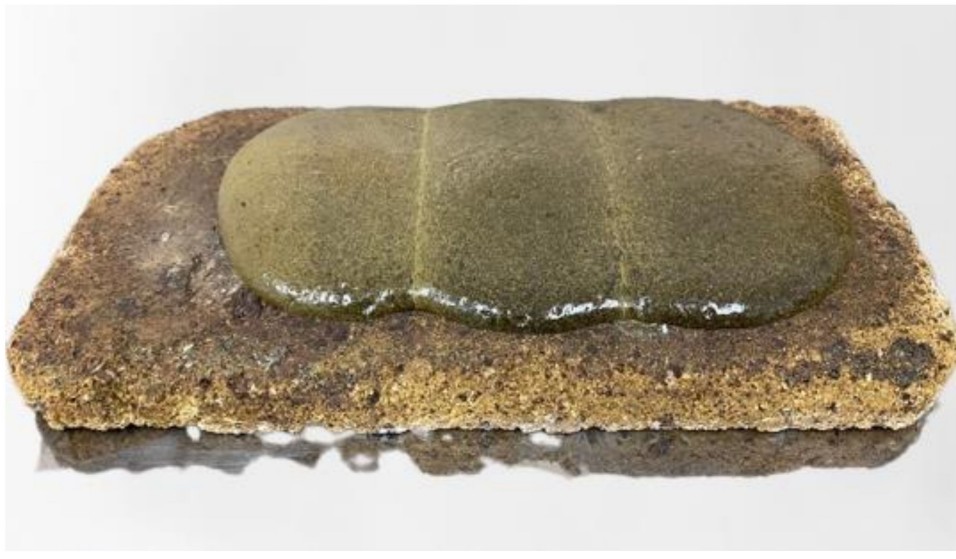

**Figure 3.** Samples with the addition of diatomite from the Utesai deposit after burning at a temperature of 1200 °C.

According to GOST 530-2012 "Ceramic brick and stone. General technical regulations", the obtained samples with the addition of diatomites correspond to the second class of medium density. According to the compressive strength, this is M 400 and M 500-clinker brick. However, the authors of [5] showed the possibility of obtaining grade M 800 and M 1000 clinker wall products. Such high strength grades are due to the high cristobalite and glass phase contents in the products, the formation of which is due to the addition of up to 30% diatomite. In our case, with the addition of 30% diatomite (Table 1, composition No.4) and burning at 1100 °C, a sample with a compressive strength of 13.27 MPa or grade M 125 was obtained. The authors of [5] used fusible clay and diatomite, which is significantly different from ours regarding the chemical and mineralogical composition, in connection with the ceramic shards of the products formed in these phases.

### 3.3. Study of the Influence of Bentonite Clay on the Properties of Products

Cylinder samples based on loam and bentonite clay, after drying, were burned in a muffle furnace at 1100 and 1150 °C for 1 h.

Table 4 shows the compositions and results of the properties of the samples after burning at 1100 °C based on loam without additives and the addition of bentonite.

**Table 4.** Compositions of the charge and properties of products with the addition of bentonite clay.

| No. | Compositions (Compound) | $\rho_{av.}$, kg/m$^3$ | Compressive Strength, MPa | Full (Total) Volumetric Shrinkage, % | Note |
|---|---|---|---|---|---|
| | | | Burning Temperature 1100 °C | | |
| **1** | Loam-100% | 1647 | 17.46 | 4.1 | Color of the sample is light yellow |
| 2 | Loam-90%; bentonite-10% | 1588 | 21.42 | 4.0 | Color of the sample is light yellow with a green tint |
| 3 | Loam-85%; bentonite-15% | 1570 | 21.46 | 3.8 | Color of the sample is light yellow with a green tint |
| 4 | Loam-80%; bentonite-20% | 1594 | 15.58 | 3.4 | Color of the sample is light yellow with a green tint |
| 5 | Loam-75%; bentonite-25% | 1604 | 11.97 | 2.9 | There is small transverse cracks on the sample |
| 6 | Loam-70%; bentonite-30% | 1555 | 9.8 | 2.1 | The specimen has transverse cracks and swelling (small) at the base. |

The addition of of 10, 15, and 20% bentonite led to a slight decrease in the average density of the samples of 3.5%, 4.7%, and 3.2%, respectively. With the addition of 10 and 15% bentonite, the compressive strength of the products increased by 22.7% and 22.9% corresponding to a compressive strength of 21.42 and 21.46 MPa (grade M 200), respectively. Some research has also shown that the addition of 15 and 20% bentonite to the blast-furnace slag contributes to the production of grade M 175 and M 200 ceramic bricks [2], and the addition of phosphate slag produced wall bricks with a compressive strength of 17.2 to 21.5 MPa or grade M 175 and M 200 [3]. This is probably due to the fact that bentonite is a fusible clay that can easily enter, during burning, solid-phase reactions with loam, blast-furnace, and phosphorus slag to form easily fusible eutectics. This leads to the formation of the melt, which improves the sinterability of the ceramic tiles and, consequently, increases the strength, due to the formation of a glass phase, after the cooling of samples.

Samples with 35, 40, 45, and 50% bentonite showed significant transverse cracks after burning. As the amount of bentonite in the samples increased, the size of the cracks became larger. Figure 4 shows photos of the samples with 35 and 40%, bentonite which show significant transverse and longitudinal cracks. In [8], the authors indicated that the addition of bentonite improves the swelling of the clay and this was the basis for the development of lightweight aggregate technology. Apparently, in our studies, the addition of 30% bentonite or more also contributes to the swelling of the ceramic weight, which was manifested in the formation of longitudinal and transverse cracks in the samples. Bentonite contains up to 70% of the mineral montmorillonite ($Al_2(OH)_2$)[$Si_4O_{10}$]·$nH_2O$), and the particle size of the mineral is less than 1 mcm. Monmorillonite swells when moistened and its volume increases 16-fold. The laminar structure of the mineral montmorillonite contains silicate layers, the maximum length between which is about 1.4 nm, that are separated by layers of water molecules, and the thickness of these layers can vary over a wide range. Therefore, drying removes physically and, during burning, chemically bound water, which, in the form of vapor, moves the laminar silicate layers vertically, which leads to the formation of transverse cracks.

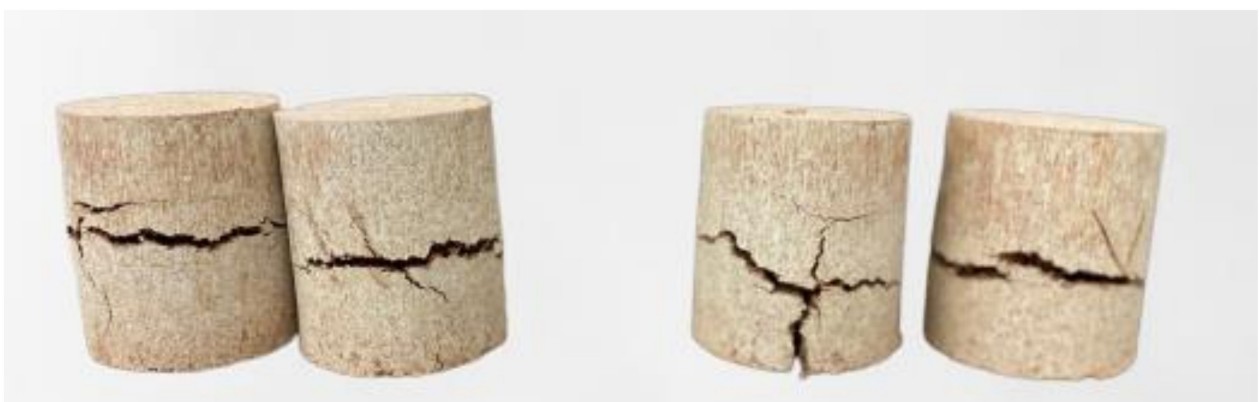

**Figure 4.** The samples with bentonite after burning at 1100 °C for 1 h.

The addition of 10, 15, and 20% bentonite slighty decreased the average density of the samples by 3.5%, 4.7%, and 3.2%, respectively. With the addition of 10 and 15% bentonite, the compressive strength increased by 22.7% and 22.9%, respectively. Samples with 20, 25, and 30% bentonite showed a strength of 10.8%, 31.4%, and 43.9% less than the pure loam sample, respectively (Table 4). The significant decrease in the strength of the samples with 25 and 30 % bentonite is associated with the formation of small longitudinal cracks (Table 4).

To determine the impact of temperature on the properties of the products, samples with 15 and 20 % bentonite were subjected to burning.

Table 5 presents the compounds and results of the properties of the samples based on loam without additives and with bentonite after burning at 1150 and 1170 °C.

**Table 5.** The compounds and properties of samples with bentonite.

| No. | Compounds | $\rho_{av.}$, kg/m$^3$ | | Compressive Strength, MPa | | General Shrinkage, % | |
|---|---|---|---|---|---|---|---|
| | | 1150 °C | 1170 °C | 1150 °C | 1170 °C | 1150 °C | 1170 °C |
| 1 | Loam-100% | 1638 | 1927 | 18.17 | 38.5 | 5.2 | 6.0 |
| 2 | Loam-85%; bentonite-15% | 1575 | 1737 | 21.71 | 42.7 | 4.9 | 6.5 |
| 3 | Loam-80%; bentonite-20% | 1554 | - | 16.6 | - | 5.1 | - |

According to the data in Table 5, burning the products at 1150 °C increased the compressive strength of the pure loam sample by 4% and 6% with the addition of 20% bentonite, compared to the strength after burning at 1100 °C. The strength of the sample with the addition of 15% bentonite practically did not change. The optimal addition of bentonite clay is 15%. In this case, a decrease in the average density and an increase in the compressive strength during burning at 1100 °C were observed. Burning at 1150 °C did not lead to a significant increase in the compressive strength of the specimens. Burning of the samples with 15% bentonite at 1170 °C led to a significant increase in the compressive strength of the samples and an increase in the average density, which is related to the formation of a liquid phase and sintering of the ceramic shard (Table 5). Figure 5 shows photographs of the samples based on loam without additives and with 15% bentonite after burning at 1170 °C.

Figure 5 shows that after burning at 1170 °C, the samples showed slightly vitrified surfaces, which indicates the sintered state of the ceramic shard. According to Table 5, the loam-based sample without additive had a compressive strength of 38.5 MPa and an average density of 1927 kg/m$^3$. The addition of 15% bentonite contributed to the production of products burned at the same temperature, with a compressive strength of 42.7 MPa and an average density of 1737 kg/m$^3$.

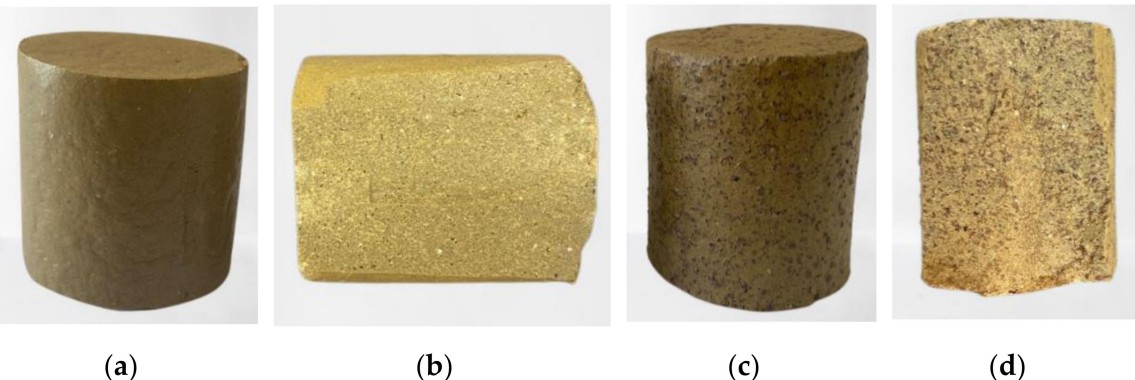

<center>(**a**)       (**b**)       (**c**)       (**d**)</center>

**Figure 5.** Samples after burning at a temperature of 1170 °C: (**a**) sample based on loam without additive. (**b**) Structure of the sample after the compression test. (**c**) Sample with bentonite. (**d**) Structure of the sample after the compression test.

According to GOST 530-2012, the samples obtained with the addition of bentonite clay correspond to class 2 with a medium density. After burning at 1100 °C, in terms of compressive strength, they correspond to grade M 200, and when burned at 1170 °C, they correspond to clinker bricks (grade M 400).

Thus, it was found that the addition of diatomite from the Zhalpak and Utesai deposits to loam decreased the average density of the samples burned at 1000 and 1100 °C and increased their compressive strength. At the same time, the optimal addition of diatomite was found to be 15%, with a fractional composition of 0.315–0.16 mm. The addition of bentonite clay, in general, reduced the strength and density of the samples. The optimal addition of bentonite was 15%, which helped to reduce the average density and increase the compressive strength of the products after burning at a temperature of 1100 °C. It was found that the optimal addition of diatomite and bentonite clay of 15% to loam make it possible to obtain, after burning at a temperature of 1170 °C, ceramic products that have the same average density and compressive strength as clinker bricks.

### 3.4. Study of the Structure and Phase Composition of Ceramic Samples

To determine the phase composition, X-ray diffraction analysis of the samples obtained based on pure loam and with additives was carried out. Figure 6a–d show the diffraction patterns of the samples after burning at a temperature of 1170 °C.

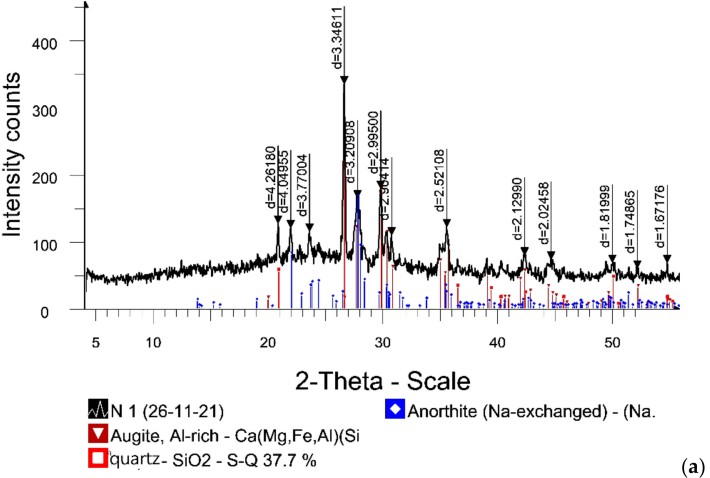

**Figure 6.** *Cont*.

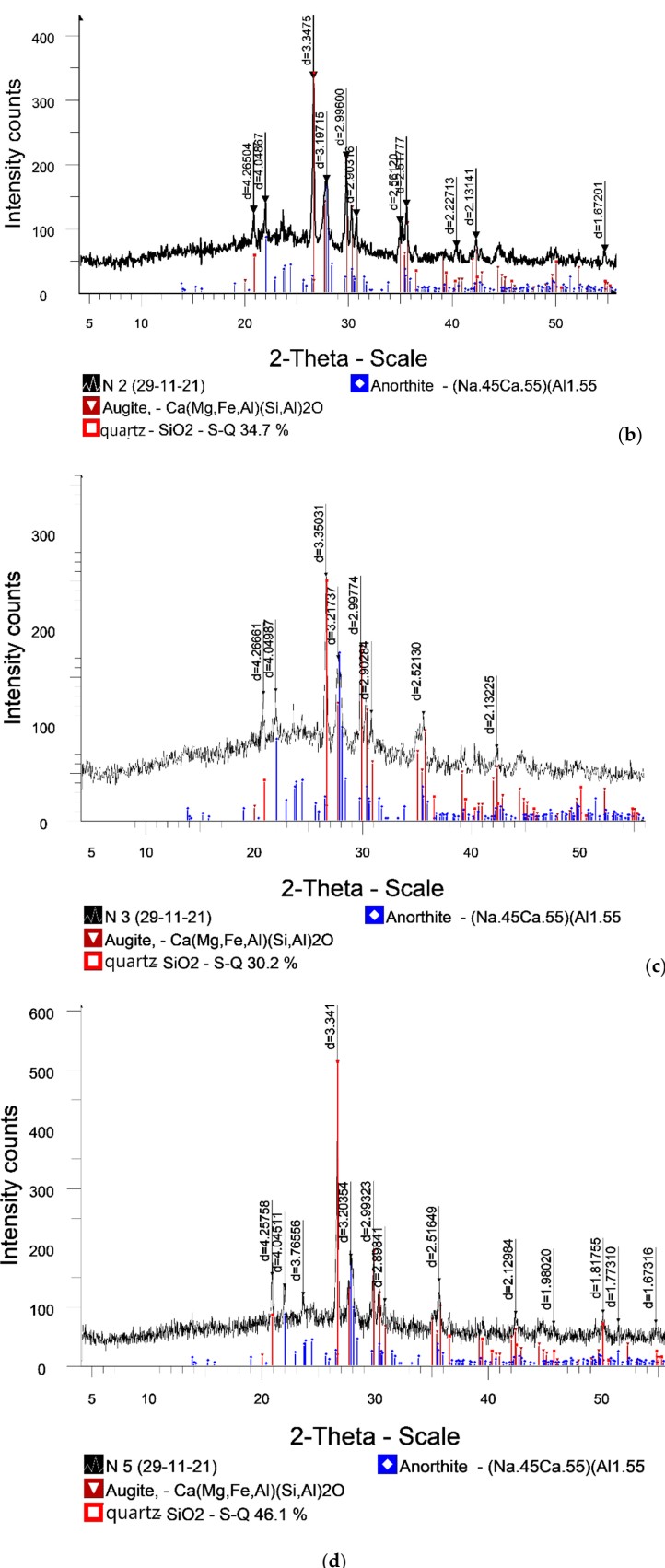

**Figure 6.** Diffractograms of loam-based samples after burning at 1170 °C: (**a**) without additives. (**b**) With the addition of 15% diatomite from Zhalpak deposit. (**c**) With the addition of 15% diatomite from Utesai deposit. (**d**) With the addition of 15% bentonite clay.

All of the above diffraction peaks belong to the above phases. The results of the semi-quantitative X-ray diffraction analysis are shown in Table 6.

**Table 6.** The phase compositions of samples burned at 1170 °C.

| Compositions | Name of Mineral Phases and Their Amounts | | |
| --- | --- | --- | --- |
| | **Augite** | **Quartz** | **Anorthite** |
| Loam-100% | 49.2% | 37.7% | 13.1% |
| Loam-85%; diatomite from Zhalpak deposit-15% | 52.7% | 34.7% | 12.6% |
| Loam-85%; diatomite from Utesai deposit-15% | 54.7% | 30.2% | 15.1% |
| Loam-85%; bentonite clay-15% | 42.7% | 46.1% | 11.2% |

The interpretation of the diffractograms and semi-quantitative X-ray phase analysis indicated that augite was present as the main crystalline phase of the samples obtained from pure loam and with the addition of diatomite. In the sample with the addition of bentonite clay, the predominant phase was quartz. In all samples, anorthite was present in small amounts. The addition of diatomite contributed to the formation of a large amount of augite in the composition of the products, which led to increased strength. Although the sample based on loam without additives is dominated by augite, its compressive strength was found to be lower than the strength of the sample (Table 5) with the addition of bentonite clay, which was dominated by quartz (Figure 6d). Probably, the addition of bentonite clay improved the sintering ability of the ceramic shards due to the formation of a large volume of the glass phase, thereby increasing its strength.

Figure 7 shows the structures of the samples after burning at 1170 °C.

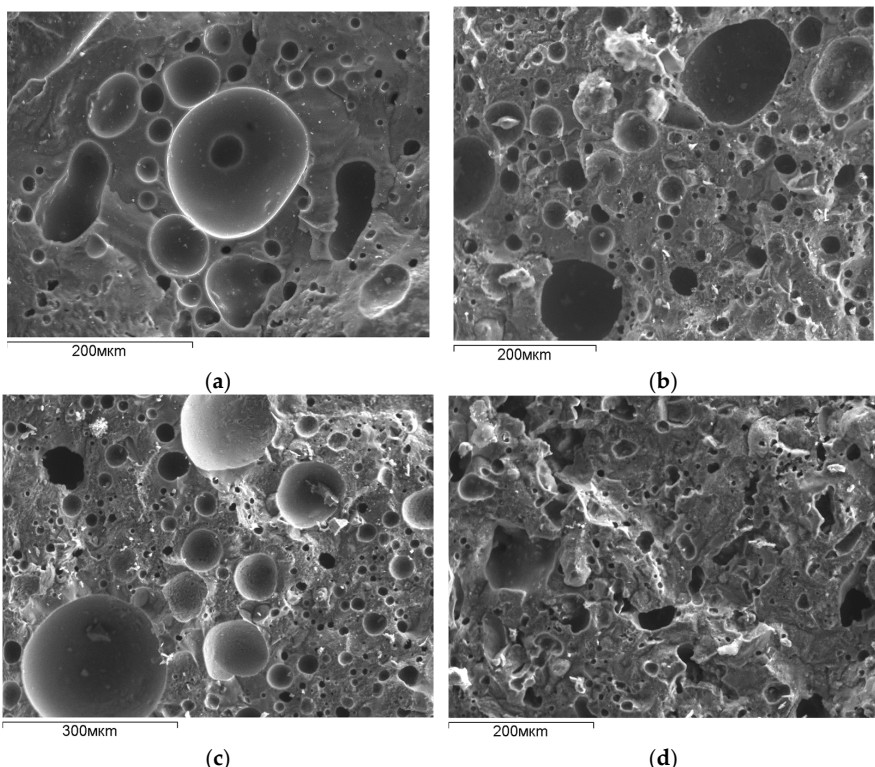

**Figure 7.** Electron microscopic photographs of the structure of samples after burning at 1170 °C: (**a**) sample based on loam without additives. (**b**) Sample based on loam with 15% diatomite from the Zhalpak deposit. (**c**) Sample based on loam with 15% diatomite from the Utesai deposit. (**d**) Sample based on loam with 15% bentonite.

The study of the structure of the samples after burning showed (Figure 7) the presence of a significant number of pores. In this case, the structure of the sample based on loam without additives is characterized by large pore sizes and a small amount of the glass phase (Figure 7a). The addition of diatomite led to the formation of a greater glass phase (Figure 7b,c-"balls") and a decrease in the pore size. This is probably why the samples with diatomite additives are more amorphous. The structure of the sample with bentonite also showed the presence of a glass phase (Figure 7d-"balls"). However, the amount is less compared to the samples with diatomite additives.

## 4. Conclusions

1. We carried out research on the efficiency of diatomite and bentonite on the characteristics of local loam, aiming to develop ceramic masses for the manufacturing of bricks with improved physical and mechanical characteristics.

2. It was found that diatomite additives from the Zhalpak deposit decrease the average density of products and increase their compressive strength. Ceramic masses with the addition of 15% to 30% diatomite can be used to produce conditionally efficient wall materials with an average density ranging from 1410 to 1550 kg/m$^3$ and a compressive strength of 10 to 14 MPa, after burning in the temperature range from 1000 to 1100 °C.

3. Regarding the influence of diatomite from the Utesai deposit, the following fractions were investigated: 2.5–1.25, 1.25–0.63, 0.63–0.315, and 0.315–0.16 mm, in the quantity of 15%. The optimal fraction was 0.315–0.16 mm, which allowed a sample with an average density of 1521 kg/m$^3$ and a compressive strength of 14.8 MPa to be obtained after firing at 1100 °C. After burning at 1150 °C, the samples had sintered porous dense tiles, and their compressive strength was 29.4 MPa.

4. It was found that the addition of bentonite, in general, reduced the strength and density of the samples. The optimal addition of bentonite is 15%, which decreased the average density and increased the compressive strength of the samples after burning at 1100 °C.

5. It was found that the optimal additions of 15% diatomite and bentonite clay to clay loam allowed yo high-strength ceramic products to be obtained after burning at a temperature of 1170 °C. According to GOST 530-2012 "Bricks and Ceramic Stone", the samples correspond to the class 2 average density. In terms of the compressive strength, this corresponds to grades M 400 and M 500. In terms of the average density and compressive strength, ceramic samples meet the requirements for clinker bricks.

6. Using X-ray diffractometric analysis, it was established that augite prevails as the main crystalline phase in the samples obtained on the basis of loam with the addition of diatomite. In the sample with the addition of bentonite clay, the predominant phase is quartz. Anorthite is present in small amounts in all samples. Diatomite additives contribute to the formation of a large amount of augite in the composition of the products, which leads to higher strength indicators.

7. The study of the structure of the samples after burning confirmed the presence of a significant number of pores. In this case, the structure of the sample based on loam without additives is characterized by large pore sizes and a small amount of the glass phase. The addition of diatomite leads to the formation of a greater glass phase and a decrease in the pore size. The structure of the sample with bentonite shows the presence of a smaller glass phase compared to the samples with the addition of diatomite.

**Author Contributions:** Conceptualization, R.E.N. and Z.O.Z.; methodology, Z.O.Z. and Y.S.O.; software, A.A.M.; validation, A.A.J., Z.O.Z. and R.E.N.; formal analysis, M.T.Z. and Y.S.O.; investigation, M.T.Z. and R.E.N.; resources, R.E.N.; data curation, R.E.N. and A.A.J.; writing—original draft preparation, Z.O.Z.; writing—Z.O.Z. and M.T.Z.; visualization, R.E.N. and A.A.M.; supervision, R.E.N.; project administration, R.E.N.; funding acquisition, R.E.N. All authors have read and agreed to the published version of the manuscript.

**Funding:** This research is funded by the Science Committee of the Ministry of Education and Science of the Republic of Kazakhstan and was done as part of the project AP09058365 "Development of technology for high-strength clinker and energy-efficient ceramic wall products from clay and opal-cristobalite rocks" in the framework of "Grant funding for young scientists for scientific and (or) scientific and technical projects for 2021–2023".

**Institutional Review Board Statement:** Not applicable.

**Informed Consent Statement:** Not applicable.

**Data Availability Statement:** Not applicable.

**Acknowledgments:** The authors are grateful to the leaderships of the LLP SAVENERGY, Satbayev University and the Science Committee of the Ministry of Education and Science of the Republic of Kazakhstan for creating the conditions for carrying out this work.

**Conflicts of Interest:** All authors declare that there are no conflict of interest.

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
