# Peer review of "An Investigation of the Effects of Additives and Burning Temperature on the Properties of Products Based on Loam"

_applsci, doi:10.3390/app12073352_

Round 1

Reviewer 1 Report

The manuscript “Study of Additives Effects and Burning Temperature on the Properties of Products Based on Loam” by Nurlybaev R.E., Zhuginissov M.T. et al. is interesting and presents a solution to a problem in the field of the ceramic technology. The authors presented study on development the composition of the ceramic mass for the manufacture of bricks with improved properties. They used local loam from the Almaty deposit in Kazakhstan and tested influence of additives of diatomite or bentonite clays for the parameters of the obtained ceramic samples. On the basis of the performed measurements, it was determined what the percentage share of additives in the raw material composition should be optimal, its fractional composition (obviously the finest) and proper temperature of sintering. The authors correctly solved the technological problem. The idea of using diatomite and bentonite turned out to be effective. Obtained samples of the bricks material corresponded to the 2nd class in terms of average density and compressive strength grades M 400 and M 500. The assumptions were therefore achieved successfully.

From a scientific point of view, the text is basically correct, explicit, has a message, and can be of interest to the professional readers. It contains only insignificant defects and does not require more substantial changes. The manuscript is not a simple continuation of earlier works of the authors, with only a minor improvement. The work is performed in an adequate manner. Presented investigation is of sufficient importance to motivate publication in Applied Sciences. In my opinion the paper is accessible to the general readership of the journal. The background and conceptual framework are satisfactory. The title of this paper appropriately reflects the purpose of the study. An abstract indicates the experimental approach and methods. In the first chapter, Introduction, the same text was repeated twice - on lines 28-174 and then again on lines 175-321. In consequence this chapter is very long – 7 pages. It is surprising that the authors did not bother to read the introduction to the manuscript before sending the work to the editor. The purpose of the next section Materials and methods was to provide a general introduction a concept of interest and to present testing methods, as well as test procedure. The experimental section (Results and discussion) is generally properly written. Conclusions are clear and adequate, references (18) – sufficient. Thus the manuscript can be published after minor revision.

  • The repetition of the text should be removed from the introduction - then it will be of the appropriate length.
  • In my opinion it is pointless to include school equations 1 and 2. It can be written that density and compressive strength were determined in a typical way.

From a scientific point of view, the work is not revealing. Bentonite is often used in raw material compositions - it improves plasticity and facilitate sintering. The use of the finest possible grain fractions is obvious.

Author Response

Response to Reviewer 1 Comments

Dear Reviewer!

Thank you for your consideration and appreciation of our manuscript “Study of Additives Effects and Burning Temperature on the Properties of Products Based on Loam”. 

Point 1: In the first chapter, Introduction, the same text was repeated twice - on lines 28-174 and then again on lines 175-321. In consequence this chapter is very long – 7 pages. It is surprising that the authors did not bother to read the introduction to the manuscript before sending the work to the editor.

Response 1: Yes, we agree, it is repetitive. Shortened, removed repetitive text on lines 175-321.

Point 2: The purpose of the next section Materials and methods was to provide a general introduction a concept of interest and to present testing methods, as well as test procedure. The experimental section (Results and discussion) is generally properly written. Conclusions are clear and adequate, references (18) – sufficient. Thus the manuscript can be published after minor revision.

The repetition of the text should be removed from the introduction - then it will be of the appropriate length.

In my opinion it is pointless to include school equations 1 and 2. It can be written that density and compressive strength were determined in a typical way.

Response 2: Yes, we agree, we removed those equations.

Point 3: From a scientific point of view, the work is not revealing. Bentonite is often used in raw material compositions - it improves plasticity and facilitate sintering. The use of the finest possible grain fractions is obvious.

Response 3: The scientific significance of the study consists in determining the optimal compositions in the percentage ratio of diatomite and bentonite in ceramic products and their impact on its properties. During the study, the amount of diatomite and bentonite in the composition of the products, as well as the sintering temperature of the samples were varied. The study showed that the optimal addition of diatomite and bentonite to loam reduces the average density of products and increases their strength characteristics.

Reviewer 2 Report

The authors investigated the Additives Effects and Burning Temperature on the Properties of Products Based on Loam. This is important approach of research related to ceramic and fired bricks. The paper is suitable to applied sciences, but major revision is required.

Here are comments that could help to improve the quality of the text.

General comments

  1. Introduction
  • The introduction represents about 30% of the manuscript, and it is very long. I suggest summarizing the results of the various studies in a table and shortening this section to three or four pages.
  1. Objectives
  • Please end the introduction by stating the research aims and its novelty.
  • Materials and Methods
  • You can organize the samples used in the research in a table showing a unique ID for each series.
  1. Titles of section 3
  • Please shorten all titles of the sections: 3.1, 3.2, and 3.3
  1. Conclusions
  • Please write a summary of your results without details about the objectives, results, or discussion. Just present the new findings in this section.

Specific comments

  1. Line 16: “ Diatomite’s from the Zhalpak and Utesai deposits ….”
  • Diatomite or Diamtomites (Please, correct throughout the text)
  1. Line 43: “The author of [1] ….”
  • Write the name of the author.
  1. Line 93: “. …. average density of 1550...1700 kg/m3”
  • As “average” you should mention one value, or you can replace “average” by “varies from … to …”
  1. Line 96: “The resulting ceramic samples have a density of 1710 (1700) kg/m3, compressive strength of 22.1 (15.8) kg/m3, and water absorption of 9.8 (13.9)%, respectively.”
  • Why there are two values for each property?
  1. Line 97: “compressive strength of 22.1 (15.8) kg/m3,..”
  • Incorrect unit
  1. Line 404: section title
  • Please shorten the title to “3.1 physical and mechanical properties ….”
  1. Line 409: table 1
  • Data and results: You need first to add another table showing the composition of each series with a unique ID, then you can use the sample’s ID in this table to show the physical results of each series.
  • Why the shrinkage of samples 4,5 and 6 is not presented in spite of the changes in bulk density.
  • Explain the rule of the two major factors behind the variations in bulk density: shrinkage and partial evaporation of precursors.
  1. Table1: Sample #3:
  • Can you explain why the shrinkage decreases with increasing the heating temperature from 1000 °C to 1100 °C?.
  • 9 instead of 14,9.
  1. Line 427
  • Shorten the title
  1. Line 443: “So 443 the addition of fractions of 2.5-1.25 mm and fractions of 1.25-0.63 mm significantly 444 reduces the compressive strength of the samples in comparison with samples on pure 445 loam by 10.86 MPa and 13.96 MPa, respectively …”
  • Please use percentage instead of the values. For example, the compressive strength of 2.5-1.25 sample was reduced by 80% with addition of 15% …. Please, correct that throughout the text
  1. Section 3.3

Please combine the XRD patterns of each process in one figure to compare the phase changes due to the variations in composition or temperatures.

Author Response

Response to Reviewer 2 Comments

Dear Reviewer!

Thank you for your consideration and appreciation of our manuscript “Study of Additives Effects and Burning Temperature on the Properties of Products Based on Loam”. 

Point 1: Introduction. The introduction represents about 30% of the manuscript, and it is very long. I suggest summarizing the results of the various studies in a table and shortening this section to three or four pages.

Response 1: Yes, we agree, the introduction was shortened. We removed the repetitive text.

Point 2: Objectives. Please end the introduction by stating the research aims and its novelty.

Response 2: The introduction ended with a statement of the purpose of the research and added novelty.

Point 3: Materials and Methods. You can organize the samples used in the research in a table showing a unique ID for each series.

Response 3: It is not possible for us to specify such a ID for each series.

Point 4: Titles of section 3. Please shorten all titles of the sections: 3.1, 3.2, and 3.3

Response 4: Shortened Study of the diatomite influence fromZhalpak deposit on the properties of products. Also shortened 3.2 and 3.3.

Point 5: Conclusions. Please write a summary of your results without details about the objectives, results, or discussion. Just present the new findings in this section.

Response 5: We tried as much as possible to reduce the conclusions and their number to 7.

Specific comments

Point 1: Line 16: “ Diatomite’s from the Zhalpak and Utesai deposits ….”

Diatomite or Diamtomites (Please, correct throughout the text)

Response 1: Corrected throughout to diatomite.

Point 2: Line 43: “The author of [1] ….”

Write the name of the author.

Response 2: According to the journal instructions, instead of the author's name they wrote: At work [1]

Point 3: Line 93: “. …. average density of 1550...1700 kg/m3

As “average” you should mention one value, or you can replace “average” by “varies from … to …”

Response 3: All natural and artificial stone materials have true density and average density. True density is the density of a material in an absolutely dense state, that is, without pores. Average density is the density of the material in its natural state with voids and pores. In our article we are talking about average density, and according to the results of other authors and our studies, the average density varies within some limits and we cannot give one value. That's why we write the average density of 1550...1700 kg/m3 rather than their average value of 1625 kg/m3.

Point 4: Line 96: “The resulting ceramic samples have a density of 1710 (1700) kg/m3, compressive strength of 22.1 (15.8) kg/m3, and water absorption of 9.8 (13.9)%, respectively.”

Why there are two values for each property?

Response 4: Because the first value was obtained by the introduction of volcanic ash with a particle size of 1 mm for the clay material of montmorillonite composition - 1710 kg/m3 , the second value (in parentheses) was obtained by the introduction of volcanic ash in the clay material of polymineral composition (1700 kg/m3). This also applies to compressive strength and water absorption.

Point 5: Line 97: “compressive strength of 22.1 (15.8) kg/m3,..”

Incorrect unit

Response 5: Agreed, we corrected for 22.1 (15.8) MPa.

Point 6: Line 404: section title

Please shorten the title to “3.1 physical and mechanical properties ….”

Response 6: Shortened 3.1 Study of the influence of diatomite of Zhalpak deposit on the properties of products. Also reduced 3.2 and 3.3.

Point 7: Line 409: table 1

Data and results: You need first to add another table showing the composition of each series with a unique ID, then you can use the sample’s ID in this table to show the physical results of each series.

Why the shrinkage of samples 4,5 and 6 is not presented in spite of the changes in bulk density.

Explain the rule of the two major factors behind the variations in bulk density: shrinkage and partial evaporation of precursors.

Response 7: It is not possible for us to specify such a ID.

The shrinkage of samples 4,5 and 6 is not presented, due to the fact that the compressive strength of the samples began to decline and they were no longer of practical interest.

Shrinkage and evaporation of precursors occurs as a result of the general impact of all physical and chemical processes occurring in samples with added diatomite and concretite during firing - dehydration of minerals, material transfer due to diffusion, recrystallization and formation of new minerals, as well as due to the convergence of particles under the action of surface tension forces of the liquid phase during sintering.

Point 8: Table1: Sample #3:

Can you explain why the shrinkage decreases with increasing the heating temperature from 1000 °C to 1100 °C?.

9 instead of 14,9.

Response 8: Shrinkage values: at 1000 oC shrinkage is 4.3%, at 1100 oC shrinkage is 4.1%. At these shrinkage values, the strength is 14.9 and 13.24 MPa, respectively. It is likely that the introduction of 20% or more diatomite in the composition of loam, and firing at 1100 oC are noticeable modification of quartz contained in the diatomite. Modification transformations of quartz are accompanied by an increase in quartz volume and the formation of internal stresses, which lead to a decrease in strength. This can be clearly seen in the reduction of strength of samples No. 4, No. 5 and No. 6.

9 instead of 14,9 - It is not clear where these shrinkage values are taken? 14.9 is the value of compressive strength of sample #3 at 1000 oC.

Point 9: Line 427

Shorten the title

Response 9: All titles were shortened in section 3: 3.1, 3.2 и 3.3

Point 10: Line 443: “So 443 the addition of fractions of 2.5-1.25 mm and fractions of 1.25-0.63 mm significantly 444 reduces the compressive strength of the samples in comparison with samples on pure 445 loam by 10.86 MPa and 13.96 MPa, respectively …”

Please use percentage instead of the values. For example, the compressive strength of 2.5-1.25 sample was reduced by 80% with addition of 15% …. Please, correct that throughout the text

Response 10: Corrected throughout the text for percentages.

Point 11: Section 3.3

Please combine the XRD patterns of each process in one figure to compare the phase changes due to the variations in composition or temperatures.

Response 11: All the radiographs were combined into one table as a comparison. Our software does not allow combining radiographs in one figure.

Reviewer 3 Report

The English of the manuscript is not acceptable as it contains many English errors.
Introduction is too much long: 7 pages. Authors should propose an overview. The novelty of the study is not clear. The gap in the literature is not clear. 
We don’t really understand why this research is necessary and is significant.
All the figures (and tables) are poor quality (with the green background !)
In the materials and methods section, there is no sub-parts. Some equations are obvious : Do you really think it is necessary to detail how to calculate a density ?? Or a compressive strength ?
In the results and discussion section, the authors did not re-quote the references used in the introduction once. The discussion part should however compare your results to previous studies.
This paper is unfortunately more of a test report than a scientific article. The analysis of the results and their explanation is not sufficiently developed.This is also clearly visible in the conclusion. Indeed, this is only a summary of results. No explanations are given. Finally, no perspective for this work is offered by the authors. Discussion and conclusion sections should be improved.
Overall, the paper, at its current condition has several serious issues, which makes it not acceptable for publication.

Author Response

Response to Reviewer 3 Comments

Dear Reviewer!

Thank you for your consideration and appreciation of our manuscript “Study of Additives Effects and Burning Temperature on the Properties of Products Based on Loam”. 

Point 1: The English of the manuscript is not acceptable as it contains many English errors.

Response 1: We'll refine the English.

Point 2: Introduction is too much long: 7 pages. Authors should propose an overview. The novelty of the study is not clear. The gap in the literature is not clear. We don’t really understand why this research is necessary and is significant.

Response 2: The introduction was shortened, ended with a statement of research objectives (purpose) and added novelty of study.

The aim of the study is to develop compositions of ceramic mass for the manufacture of bricks with improved properties based on local loam with the use of diatomite and bentonite clay.

Novelty of the work: it was found that the addition of diatomite and bentonite clay to clay loam increases the strength of products at a decrease in their average density due to the formation of glass phase, formed during sintering of ceramic shards; It was found that the addition of diatomite leads to the formation of more augite, and the addition of bentonite clay increases the content of quartz in the composition of products; it is shown that products with an increased content of augite have higher strength characteristics.

.

Point 3: In the materials and methods section, there is no sub-parts. Some equations are obvious : Do you really think it is necessary to detail how to calculate a density ?? Or a compressive strength ?

Response 3: We added subsections to the Materials and Methods section. Yes, we agree, we removed those equations.

Point 4: In the results and discussion section, the authors did not re-quote the references used in the introduction once. The analysis of the results and their explanation is not sufficiently developed.This is also clearly visible in the conclusion. Indeed, this is only a summary of results. No explanations are given. Finally, no perspective for this work is offered by the authors. Discussion and conclusion sections should be improved.
Overall, the paper, at its current condition has several serious issues, which makes it not acceptable for publication.

Response 3: In the Results and Discussion section, we presented the results of our own work and discuss the results obtained from our own research, so we do not cite other authors here. According to the review of other reviewers more asked to reduce the results, it was suggested to present only the main results and reduce the detailed data in the results and conclusion section.

Point 5: Overall, the paper, at its current condition has several serious issues, which makes it not acceptable for publication.

This is our first time submitting to your journal and we are learning all the requirements for submitting manuscript on the AS journal platform. We ask you to review our work as much as possible and we will finalize anything that needs to be corrected. In our city often cuts power due to the current emergency situation, despite this we will try to work on our article to solve the issues on reviewers feedback. 

Round 2

Reviewer 2 Report

The Authors have addressed all of my concerns with the original manuscript.

Author Response

Dear Reviewer,

Thank you very much for your work and for your valuable advice in writing the manuscript.

Best Regards,

Author's team

Reviewer 3 Report

Thank you for sent answers. Some specific answers to the specific comments are satisfactory, but the general remarks have been omitted.

So please take into consideration these remarks :

  • Please improve the quality of Figures : delete the green background and DO NOT deform the figures, respect the orginal height/width ratio (ex : Fig. 6 and 7 are deformed).
  • In the results and discussion section: you cannot just present your results. You have to analyse, make hypothese  and/or link with the observations made in the other papers in your litterature review. If you just present your results (with a description), this is a technical paper, not a scientific paper!

Author Response

Dear Reviewer,

Thank you very much for your consultation and support of our manuscript. You have guided us very well to improve our work.

All your comments have been improved. We are improved the all figures and "Results and discussion" section (p.p.7, 10, 11, 12). 

We are analyzed and linked with the observations made in the other papers in our literature review. 

Thank you.

Best Regards,

Authors team